# The Cardioprotective Properties of Pulses and the Molecular Mechanisms of Their Action

**DOI:** 10.3390/ijms26051820

**Published:** 2025-02-20

**Authors:** Beata Olas

**Affiliations:** Department of General Biochemistry, Faculty of Biology and Environmental Protection, University of Lodz, Pomorska 141/3, 90-236 Lodz, Poland; beata.olas@biol.uni.lodz.pl; Tel./Fax: +48-42-635-4485

**Keywords:** cardiovascular disease, legumes, pulses, seeds

## Abstract

Legumes and their seeds (pulses) have various nutritional and health benefits and form an important part of a healthy diet. The present work reviews recent studies from the literature concerning the cardioprotective properties of legumes, particularly pulses, and summarizes their molecular basis. The literature was gathered from electronic databases, including ScienceDirect, PubMed, SCOPUS, Web of Knowledge, Sci Finder, and Web of Science, using the following terms: “legume seeds”, “pulses”, “beans”, “peas”, “broad beans”, “chickpea”, “lentils”, “cardiovascular system”, and “cardiovascular disease”, and their combinations. The abstracts of any identified articles were initially analyzed to confirm whether they met the inclusion criteria. Pulses may reduce the risk of cardiovascular diseases (CVDs) by various mechanisms, including altering plasma lipid composition (especially lowering total and LDL cholesterol), increasing satiety, reducing inflammation, and decreasing oxidative stress and blood pressure. It is, however, unclear whether pulses maintain their cardioprotective properties after processing, and more research is needed in this area.

## 1. Introduction

Legumes reside within the Leguminosae family (also called Fabaceae), the third largest family of angiosperms, containing about 20,000 species and 750 genera. The family comprises three subfamilies: Mimosoideae, Faboideae, and Caesalphinioidae. The greatest variety of legumes can be found in tropical and subtropical regions. The fruit, known as pods, is edible together with its seeds, or pulses. Despite their variety, relatively few legume species have been used for food. The United Nations Food and Agriculture Organization divides legumes into eleven classes, as presented in Figure 1 [1,2], while McCrory et al. [3] propose a fourfold classification consisting of pulses, sow crops, oil seeds, and vegetable crops.

Pulses play an important role in traditional diets. For example, they have featured prominently in the diets of developing countries as economical alternatives to animal protein and are often used in folk medicine [4,5,6,7]. The mean global consumption of pulses is about 21 g per capita per day [8]. However, some regions, such as Europe, East Asia, Caucasus, and Central Asia, have a very low consumption. Australian children (aged two to sixteen) also consume only 4 g to 12 g of pulses per day [9]. To address this, the United Nations Food and Agriculture Organization have promoted the culinary and agricultural uses of pulses, and declared 2016 the International Year of Pulses [8].

The most commonly known and frequently consumed pulses worldwide are lentils (*Lens culinaris* L.), peas (*Pisum sativum* L.), chickpeas (*Cicer arietinum* L.), and beans (*Phaseolus vulgaris*); together, their production constitutes approximately 70% of the estimated total global output of pulses: lentils (7%), beans (30%), peas (15%), and chickpeas (17%) [10]. However, their consumption is rather low, i.e., below 3.5 kg per capita per year [2]. This low consumption has been attributed to gastrointestinal problems (as pulses may cause flatulence, cramps, and diarrhea), as well as the prolonged cooking that pulses often require, which can take over 70 min [7,11,12]—although lentils only require about 25 min to cook, making them convenient for human consumption [13]. Despite this, in developed countries, pulses are utilized more for animal feed [14].

The role of plant and fruit seeds in the human diet has increased considerably, especially in Western countries. Seeds are known to contain high levels of bioactive phytochemicals with antioxidant and anti-platelet properties, which are beneficial in inter alia cardiovascular diseases (CVDs) [15,16]. As such, incorporating edible seeds into one’s daily diet could be an important step toward ensuring good health.

Being abundant in bioactive compounds, pulses play a vital role in human nutrition and wellness. Pulses can be also used as functional food and are added to a wide range of dishes to improve their nutritional content and aid with certain health issues. They are also good sources of nutrients, including protein (usually 21–26%), especially amino acids [17], as well as dietary fiber, minerals, vitamins (especially water-soluble vitamins), and low-glycemic-index carbohydrates [5,18,19,20].

Various reports indicate that the peptides of legume seeds have cardioprotective properties, with consumption associated with lowering cholesterol, reducing blood pressure, and exerting antithrombotic effects [21,22]. Pulses contain also various antioxidant compounds, including phenolic compounds and saponins [5,9,22,23,24,25,26,27,28,29,30].

Various studies have also associated the regular consumption of pulses with various health benefits [31,32,33,34,35,36,37]. For example, a systematic review of randomized controlled trials found that a diet rich in pulses has beneficial effects on human health, indicating a daily consumption of 54–360 g cooked pulses to have cardioprotective benefits [38,39].

Although a number of review papers indicate that legumes and their pulses have health benefits [2,5,7,13,40], these present little information about the key ingredients with cardioprotective potential or the molecular mechanisms of their action [2,5,7,13]. For example, the review by Padhi and Dan Ramadath [10] only describes the relationship between pulse consumption and reductions in cardiovascular disease risk factors. The present review not only provides an overview of the cardioprotective potential of pulses, but also indicates which of their components have cardioprotective potential and summarizes the molecular mechanisms behind them.

## 2. Research Methods

ScienceDirect, PubMed, SCOPUS, Web of Knowledge, Sci Finder, and Web of Science were searched for papers examining functional ingredients with cardioprotective potential present in pulses. The following search terms were used: “legume seeds”, “pulses”, “beans”, “peas”, “broad beans”, “chickpea”, “lentils”, “cardiovascular system”, and “cardiovascular disease”, or combinations of these terms. No time criteria were applied to the search, but recent papers were evaluated first. The last search was run on 24 January 2025. The identified articles were first screened by reading the abstract. The molecular mechanisms underpinning the cardioprotective action of pulses were also analyzed and discussed as part of the search.

## 3. Nutritive and Non-Nutritive Compounds of Legumes and Their Seeds

Legumes and their seeds are characterized as having high nutritional value [5,6,10]. The main components of the most commonly known and frequently consumed pulses are presented in Table 1.

A key benefit of legumes is that they typically have a higher protein content (from 15.5 to 35%) than other cultivated plants, although this depends on the growing conditions, as well as the type and maturity of the fruit [5,6,10]. Kamboj and Nanda [42] report that the protein content of legumes mostly ranges from 196 mg/g to 360 mg/g, with black beans having particularly high values (882 mg/g). In fact, black beans have a higher protein content than milk, eggs, and meat.

Some legumes and their pulses, including the field bean (14 mg/g), moth bean (11 mg/g), and white bean (13 mg/g), are low in fat content [42,43]. Ryan et al. [44] report the total fat of peas to be about 1.4 g/100 g, consisting of 16.7% saturated fatty acids, 23.7% monounsaturated fatty acids, and 58.8% polyunsaturated fatty acids. They are also rich dietary sources of α–linolenic acid and phytosterols, which have cardioprotective potential; this has also been noted for soya [45]. Kalogeropoulos et al. [46] note that the predominant phytosterol is β-sitosterol (15.0–24.0 mg/100 g, in cooked pulses).

The cardioprotective action of pulses may be partially explained by their high content of fiber (the consumption of 100 g pulses per day provides 16 g fiber, which constitutes 80% of the minimum daily requirement), especially insoluble fiber, which comprises a rich diversity of components including cellulose, hemicellulose, and lignin. This decreases the level of cholesterol through various mechanisms, such as by increased bile acid/salt excretion in feces [41]. For example, the soluble non-starch polysaccharide from lentils has been found to reduce the risk of cardiovascular disease by increasing high-density lipoprotein (HDL) cholesterol and decreasing low-density lipoprotein (LDL) cholesterol [10]. In addition, due to the high water-binding capacity of dietary fiber, it increases the volume of stool, which prevents constipation. It also accelerates intestinal peristalsis and exerts a prebiotic effect through its beneficial impact on intestinal bacterial flora [6]. It is important that common beans have higher total dietary fiber contents (2–32 g/100 g) compared to lentils, dry peas, and chickpeas (18–26 g/100 g) [47].

The dominant phenolic compounds present in pulses are phenolic acids, procyanidins, and flavonoids. In addition to these antioxidant compounds, pulses also contain tocopherols and carotenoids, which may protect LDL cholesterol from oxidation. Indeed, various varieties of lentils, beans, peas, and chickpeas have all been found to have antioxidant properties in vitro [5,9,23,25,26,27,28,29,30,48,49].

Legumes also contain various saponins, i.e., chemical compounds containing steroids or triterpenoid aglycones that are partially attached to one or more oligosaccharides. These have been found to bestow cholesterol-lowering effects in vitro and in vivo [22,50,51]. The total saponin content ranges from 30 to 388 µg/g of dry weight in faba beans; 2.3 g/kg of dry weight in chickpeas, albeit with a wide range; and only 1.8 g/kg of dry weight in peas [22]. More details about saponins as modulators of the blood coagulation system, and their use in the prevention of venous thromboembolic incidents, are described by Olas et al. [52].

Pulses also contain the alkaloid glycosides vicine and convicine, both of which demonstrate antioxidant activity, with particularly high levels being found in faba bean seeds: 409–618 µg/g of dry seed weight for the former, and 36–3121 µg/g of dry seed weight for the latter [53]. Additionally, faba beans have been found to contain 1.95–2.60 mg/g dry weight of L-3, 4-dihydroxyphenylalanine (L-DOPA), a precursor of dopamine that may regulate blood pressure and help maintain cardiovascular health [54].

Legumes can contain also non-nutritive compounds, which can be divided into two main groups: (1) non-protein compounds, such as phenolic compounds, alkaloids, and phytic acid, and (2) protein compounds, including bioactive peptides, lectin, and protease inhibitors [6,55,56]. Interestingly, the non-nutritive compounds present in pulses may have both negative and positive effects on human health [57,58]. For example, various antinutritional compounds present in legumes can decrease the bioavailability of other compounds. The phytic acid contained in lentils, beans, and peas can inhibit the absorption of various minerals, including Cu^2+^, Zn^2+^, Mg^2+^, Ca^2+^, and Fe^2+/3+^, in the gastrointestinal tract by forming complexes with them. The tannins contained in common beans also reduce the absorption of various minerals in young women. The acute effects of consuming raw pulses may be linked to active lectins and include vomiting, nausea, and diarrhea [59].

Nutrient intake via the digestive tract can also be obstructed by the presence of intact cell walls in seeds. However, the microstructures of the cell wall of legumes have been demonstrated to be resistant to thermal damage [60]. More information about the nutritive and non-nutritive compounds is given in Figure 2.

The chemical composition, quality, and biological properties of pulses may be altered by the choice of domestic and industrial food processing techniques. Common processing techniques include boiling, soaking, blanching, germination, and microwave cooking [61]. The method of preparation and thermal processing influences the content of nutritive compounds and non-nutritive compounds. For example, canning and roasting has been found to reduce the total phenolic content and antioxidant properties [62], while boiling increases the carotenoid and tocopherol content in lentils [63] and chickpeas [64]. Cooking and fermentation have a beneficial effect by reducing the levels of lectins, which are increased by roasting and baking. On the other hand, the phytate content is reduced by cooking, soaking, sprouting, and fermentation [6]. The choice of processing technique can also influence the functional properties of pulse fibers. For example, lentil flours have greater amounts of α-galactoside (a beneficial prebiotic fiber) and less phytic acid after extrusion [65]. In another study, cooking was found to lead to a significant decrease in resistant starch content in cook beans. The authors of said study suggest that this reduction is attributed to the destruction of amylase inhibitors during the cooking process [66]. In addition, boiling kidney beans, chickpeas, and white beans increases their total fiber content [67]. Erba et al. [68] suggest that germination does not alter the composition of digestive and resistant starch.

Thermal processing can decrease the concentration of certain antinutritional compounds, improving the bioavailability of other food components, and release minerals from the food matrix [15]. Amoah et al. [61] report that the choice of processing technique can improve the bioavailability of nutrients in faba beans that are inhibited by antinutritional factors. More details about the thermal processing of edible seeds are described in another review paper [15], which reports that the thermal processing of edible seeds, including pulses, affects seeds in many ways. Thermal processing can increase or decrease their biological properties. For example, only one paper demonstrates that roasting faba beans (150 °C, 10–120 min) decreases their antioxidant capacity [69]. Therefore, further experimental testing is needed to identify the precise effects of thermal processing and other preparation methods on the beneficial effects of pulses, including their cardioprotective properties. Such studies should also aim to determine the optimal method, temperature, and duration of processing.

### 3.1. Meta-Analyses and Randomized Controlled Trials

Together with cancer, cardiovascular disease (CVD), such as myocardial infarction, heart failure, or stroke, is one of the most common causes of death in the European Union. Some of the most significant causes of CVD include high blood cholesterol levels, an unbalanced diet, atherosclerosis, hypertension, and a lack of physical activity, with smoking, overweight, and obesity being important risk factors [70].

Various epidemiological studies have examined the association between pulse consumption and CVD risk, with one study noting that a diet containing at least 100 g of pulses four times a week reduces the risk of development of heart diseases by 14% [71]. However, it is difficult to identify the beneficial action of individual pulses, as other studies have focused on general legume intake, which may include soy [72,73,74].

The cardioprotective function of legumes may be associated with the presence of bioactive compounds, which are thought to decrease plasma triglyceride contents and increase high-density lipoprotein (HDL) cholesterol [24,75,76,77,78]. The hypocholesterolemic and normolipidemic effects of lentils have been examined in a variety of models, including those based on rats with streptozotocin-induced diabetes and diabetic patients [24,75,76,77,78].

Various clinical studies have examined the effect of canned pulses, or those cooked from dried grains, on the risk of CVD, especially the levels of plasma lipids, such as LDL cholesterol. Shams et al. [78] report that the consumption of 50 g cooked lentils significantly decreases the level of total cholesterol in diabetic patients, while Jenkins et al. [24] note that a four-month daily consumption of 140 g of cooked dried lentils and other pulses significantly reduces total cholesterol by 25% in hyperlipidemic men. Dabai et al. [76] found lentils and other pulse diets to have hypocholesterolemic effects in Sprague Dawley rats. In addition, various meta-analyses suggest that incorporating pulses into the diet modifies the risk of CVD by various mechanisms; for example, consumption can alter the composition of the plasma lipid profile, especially by decreasing total cholesterol and LDL cholesterol [71,79,80]. Moreover, Vinarova et al. [81] found that legume saponins lowered cholesterol bioaccesibility and serum concentrations in various in vitro and in vivo models.

Other studies have compared the effect on plasma lipid levels of combining dietary pulses with other targeted interventions, for example a low-carbohydrate diet or a diet low in saturated fats. Several studies have also examined the cardioprotective action, especially the hypolipidemic action, of pulse-based flours, for example with muffins made from whole pea flour or pea hulls, and spray-dried peas, lentils, and chickpeas [12,32,34,35,36,37,48,82,83,84,85].

Moreover, a meta-analysis by Schwingshackl et al. [86] found the risk of hypertension to decrease by about 5% as the legume intake increases by about ≤70 g/day. Other meta-analyses indicate that the consumption of cooked pulses (162 g/day) significantly reduces systolic blood pressure [87], and that the consumption of legumes is associated with a lower risk of CVDs [88].

Other studies suggest that the consumption of legumes can support weight control [28]. For example, a systematic review and meta-analysis of randomized controlled trials found that a 26-day consumption of 130 g pulses per day promotes controlled weight loss; additionally, people who regularly consumed beans, including Breton-style beans, had a lower body weight, were 22% less likely to become obese, and were more likely to demonstrate lower systolic blood pressure than the controls [23]. McCrory et al. [3] report an inverse relationship between pulse consumption and body mass index (BMI), and risk of obesity. A similar effect has also been observed in other clinical experiments [89,90]. For example, bean extracts containing digestive enzyme α-amylase inhibitors modify the gut microbiota and inhibit starch digestion, leading to a significant reduction in body weight [91].

A key part in the development of CVD is played by inflammation [89]. Fortunately, various studies indicate that pulses have anti-inflammatory properties [92,93,94,95]. For example, Hartman et al. [96] report that the consumption of navy, pinto, kidney, and black beans (about 250 g/day, i.e., 1.5 cups) for four weeks reduces the levels of inflammation biomarkers, including C-reactive protein (CRP), and soluble tumor necrosis factor-a receptors I and II in men with colorectal cancer. A meta-analysis by Salehi-Abargauei et al. [93] also demonstrated a reduction in CRP after the consumption of non-soy legumes (N = 64). Additional clinical studies of the effects of pulse consumption on CVD risk factors are summarized in Table 2.

Their findings confirm that the consumption of pulses and their products has a cardioprotective effect in various models, including inter alia healthy subjects, overweight and obese subjects, and hypercholesterolemic adults. In particular, they associate consumption with changes in LDL and HDL cholesterol, as well as blood pressure. However, the systematic review and dose–response meta-analysis by Mendes et al. [98] indicates that the intake of legumes is not associated with a reduced risk of stroke. On the other hand, the authors suggest that an intake level of 400 g/week seems to provide the optimal cardiovascular benefit. In addition, a high consumption of legumes and pulses has been shown to have beneficial effects on cardiometabolic risk factors, albeit when also associated with a lower intake of ultra-processed foods and animal saturated fat [99,100]. However, more clinical trials should be performed to determine not only the safety but also the long-term cardioprotective effects of pulses.

### 3.2. Control of Blood Pressure—In Vitro

Angiotensin-converting enzyme (ACE) inhibitors are known to regulate blood pressure by modulating vasoconstriction, and hence may also play a role in treating hypertension. Arnoldi et al. [101] note that apigenin present in sweet lupine exhibits hypotensive action, which they attribute to it competing with angiotensin I for the active site of ACE. Kidney beans can also decrease blood pressure by inhibiting ACE activity. A study conducted by Jakubczyk et al. [21] showed that six peptides obtained from fermented faba bean seeds inhibit ACE. The molecular mass of the protein ranged from 6.5 to 97 kDa, and the highest concentration of peptides (4.78 mg/mL) was noted in the hydrolysate of seeds fermented for seven days at 30 °C. However, the highest ACE-inhibitory action (IC_50_—1.01 mg/mL) was observed for a peptide fraction obtained from a sample fermented for three days at 30 °C. These seeds also contain polyphenols, which also act as ACE inhibitors [102].

Sreerama et al. [103] found that phenolic extracts obtained from pulse flours, such as horse gram seeds, cowpea, and chickpea flour, inhibit ACE in vitro. For example, the IC_50_ for horse gram was 32.8 lg/mL. Yao et al. [104] also observed that lentil phenolic extracts prevented angiotensin II-stimulated hypertension in a rat model. However, more human clinical studies are needed to confirm the anti-hypertensive action of pulses.

### 3.3. Inhibition of Oxidative Stress—In Vitro

The different parts of pulses, and legumes in general, contain various phenolic compounds and peptides with antioxidant properties. Zhao et al. [49] observed significant differences in the antioxidant properties and phenolic compound content of extracts from various common legumes, including mung beans, pinto beans, lima beans, black beans, navy beans, lima beans, black kidney beans, chickpeas, red kidney beans, and lentils. In their in vitro model, the highest hydroxyl radical-scavenging capacity was noted for navy beans and black kidney beans (75.0% and 85.7%, respectively), and the highest total antioxidant activity was noted for the lentil extract (720.7 U/g). Fidrianny et al. [105] also found extracts from red kidney beans, green beans, and soybeans to have strong antioxidant properties against 2,2′-azino-bis (3-ethylbenzothiazoline-6-sulfonic acid) (ABTS) and 2,2-diphenyl-1-picrylhydrazyl (DPPH) in vitro; they also note a correlation between the total phenolic, flavonoid, and carotenoid contents of the extract and its antioxidant properties. For example, the phenolic compounds of the extract from red kidney beans were the major contributor in scavenging activities. This is in line with various other studies [12,19,106,107,108,109,110,111,112,113]. Moreover, Zuchowski et al. [113] found the crude extract and phenolic fraction isolated from aerial parts of the lentil plant, as well as various flavonoids such as quercetin and kaempferol derivatives, to have antioxidant properties in human plasma treated with hydroxyl radicals in vitro. However, the crude extract and the phenolic fraction demonstrated antioxidant properties, i.e., an inhibition of lipid peroxidation and reductions in protein carbonylation and thiol group oxidation, only at the highest used concentration (50 µg/mL).

Kluska et al. [114] report that various kaempferol derivatives isolated from the aerial parts of the lentil plant reduce DNA damage induced by etoposide in peripheral blood mononuclear cells. These findings are confirmed by other studies, which note that phenolic compounds from green lentil and red lentil extracts also demonstrate antioxidant effects [115,116,117,118].

Faba bean extracts are known to possess a range of valuable properties that are beneficial for human health; of these, the most extensively documented are their antioxidant properties. Among all fava bean extracts, the highest total phenol and flavonoid content has been noted for acetone extract, which demonstrated 86.47% free radical-scavenging activity; this is close to the value for ascorbic acid (97.36%) [119]. Other findings indicate that an extract from mature seeds of field beans may be also a promising source of antioxidants. The tested extract exerted a protective effect on human plasma lipids and proteins treated with H_2_O_2_/Fe^2+^ (a hydroxyl radical donor). It also effectively protected the DNA in peripheral blood mononuclear cells from oxidative damage. The authors attribute the observed antioxidant potential to the complex chemical composition of the extract: the phytochemical profile demonstrated a range of phenolic compounds, including catechins [120]. Choudhary and Mishra [119] also report a positive correlation between the presence of phenolic compounds, including catechins, in fava beans and their antioxidant activity; the acetone extract from fava beans showed significant free radical-scavenging activity, with an inhibition percentage of 86%. In addition, Jakubczyk et al. [21] report that peptides from fermented broad bean seeds inhibit lipoxygenase (LOX) activity in vitro.

The antioxidant properties of pulses may act through several pathways. For example, endothelial cells have demonstrated increased levels of catalase and dismutase, as well as reduced levels of malondialdehyde, a marker of lipid peroxidation [121].

Lentils also appear to have effective antioxidant properties. Carcea et al. [122] found that mice given wheat–lentil bread demonstrated greater antioxidant effects, when analyzed via FRAP assay, than those given wheat bread.

### 3.4. Modulation of Hemostasis—In Vitro

The components of legumes may also modulate hemostasis, and have an effect on the inhibitors of blood platelet activation [123]. The crude extract, phenolic fraction, quercetin, and kaempferol derivatives obtained from lentil aerial parts demonstrated anti-platelet potential in vitro, including anti-aggregatory and anti-adhesive properties. The anti-platelet properties were determined using various methods: for example, blood platelet aggregation was measured via turbidimetry in platelet-rich plasma. The tested preparations and chemical compounds were administered at 5 and 50 µg/mL.

### 3.5. Control of Body Weight—In Vitro, Animal, and Human Models

Pulse consumption may also play a role in weight control. Martinez-Villaluenga et al. [124] found some of the peptides in soybeans to inhibit fatty acid synthesis by interacting with the catalytic domain of thioesterase. Other studies report that the peptides present in fermented broad bean seeds inhibit pancreatic lipase [21]. More details regarding the anti-obesity and anti-hypertension properties of various peptides obtained from pulses have been described by Kiersnowska and Jakubczyk [125] and Benavides-Carrasco and Jarpa-Parra [126].

Various experiments in animal models also indicate that legume fiber has an anti-obesity effect [127,128,129]. For example, chickpea fiber was found to significantly reduce rat body weight by 14.54% [127], and Wang et al. [128] achieved a similar effect in mice. Wang et al. [128] observed that the consumption of ultrafine-ground pea insoluble dietary fiber increases the abundance of beneficial bacteria (Leptospirosis and Lactobacillus) while reducing the relative abundance of harmful bacteria, including Helicobacter. This successful modulation of microbiota composition increases the production of short-chain fatty acids, resulting in an augmented secretion of satiety hormones and presenting the potential for obesity intervention. In addition, Lactobacillus, through the production of bile acid hydrolase, has a notable influence on the metabolism of blood cholesterol by the liver, thereby impacting the dynamics of the bile acid cycle. It has been proposed that this may act by regulating lipid metabolism and influencing gut microbiota structure. In this mice experiment, ultrafine-ground pea insoluble dietary fiber was administered intragastrically once a day at a volume of 10 mL/kg b.w. (0.9 g pea fiber was mixed in 10 mL water) for four weeks.

Recently, the results of Lutsiv et al.’s study [130] performed on female dietary-induced obesity model mice (C57BL/6J mice) have indicated that the consumption of mixed pulses (a daily dose of at least 300 g of cooked pulses, for 2 weeks) may help to maximize the diversity of the gut microbiome and that the benefits of this involve the metabolism of vitamins (B_1_, B_6_, B_9_, B_12_, K_2_), amino acids (lysine, tryptophan, cysteine), short-chain fatty acids (butyrate, acetate), and other mediators of lipid metabolism. Their menu plan comprised a combination of five cooked pulses: dry beans, chickpeas, cowpeas, dry peas, and lentils.

Kadyan et al. [131] have also observed the prebiotic effects of pulse-derived resistant starches on the gut microbiome and intestinal health in aged mice colonized with human microbiota. However, improvements in analytical strategies for the identification of prebiotic saccharides from pulses are needed.

Various systematic reviews [132,133] report that different types of pulses and their ingredients can modulate the microbial population in the human gut. However, the results of the present review show that a limited number of human interventions have been carried out in this area, and these have only studied the effects of a small number of pulse types and pulse ingredients on the microbial population that inhabit the human gut. For example, some studies show that whole pulses, including chickpeas, and cooked navy bean powder can affect the abundance, diversity, and/or richness of gut microbiota in healthy adults [134] and in colorectal cancer survivors [135]. Finley et al. [32] observed the same effect after the consumption of 130 g canned pinto beans/day (for 12 weeks) in healthy adults and participants with pre-metabolic syndrome. Recently, more details regarding the gut microbiota benefits of pulse-based dietary fibers have been described by Biscarrat et al. [136].

### 3.6. Anti-Inflammatory Activity—In Vitro and Animal Models


Various in vitro studies have found pulses to exhibit anti-inflammatory activity, which is believed to involve various pathways including the inhibition of cyclooxygenase (COX) or lipoxygenase (LOX). For example, Sibul et al. [137] observed that extracts from legumes such as common beans, chickpeas, and peas can inhibit COX and LOX, although the precise nature of this activity depends on the type of legume. Also, Contreras et al. [138] found that a water–ethanol black bean extract inhibited COX and iNOS activity by 30.6% and 32.3%, respectively. Lee et al. [121] also observed lentils to induce anti-inflammatory action (a decrease in pro-inflammatory cytokines and an increase in anti-inflammatory cytokines, especially interleukin-10 (IL-10)) and renal-protective effects for ischemia–reperfusion injury in mice. Recently, Alexander et al. [89] have reported more information about the anti-inflammatory effects of lentils.

## 4. Conclusions

Numerous meta-analyses and clinical studies have demonstrated that the consumption of pulses can bestow cardioprotective benefits [39,40,48,79,80,81,82,83,84,85,86,87,88,89,90,139,140]. However, the biological mechanisms by which pulse consumption influences the cardiovascular system and CVDs in humans remains unclear and poorly defined. Even so, various chemical compounds (especially phenolic compounds, fiber, and bioactive peptides) from pulses have been found to show cardioprotective potential; these are summarized in Figure 3, together with their mechanisms. The key functional ingredients of pulses with known cardioprotective activity are given in Table 3. For example, phenolic compounds exert various effects on the body, including anti-inflammatory activity, by regulating signaling pathways such as COX, LOX, and NOS. In addition, their cardioprotective action seems to be associated with their antioxidative activity at the cellular level; they have been found to stimulate antioxidant enzymes. Phenolic compounds may also be responsible for the anti-platelet action of pulses, but their molecular mechanism needs further study.

Recently, Feng et al. [2] have described that in faba beans, L-DOPA (L-3,4-dihydroxy-phenylalanine) (amino acid, 1.95–2.60 mg/g dry weight) as a precursor to dopamine may regulate blood pressure and help maintain cardiovascular health. L-DOPA may lower blood pressure levels through its ability to stimulate the production of nitric oxide.

Future studies should also investigate the effects of pulse consumption on the human gut microbiota, an important factor in reducing the risk of CVD; while a few papers have found the fiber derived from legumes and their seeds to have advantageous effects on human health, including on dyslipidemia and obesity [41], this area generally remains poorly understood.

It is important to note that no adverse effects were found in any of subjects treated with lentils, pulses, or their products. However, more animal studies and clinical trials should be performed to determine their long-term effects and safety in vivo; the latter should aim to recruit both healthy participants and those with risk factors for CVDs, such as high total and LDL cholesterol and obesity, or those who smoke. They should also aim to clarify the effects of processing on the cardioprotective properties of pulses, as this area is currently poorly described.

## Figures and Tables

**Figure 1 ijms-26-01820-f001:**
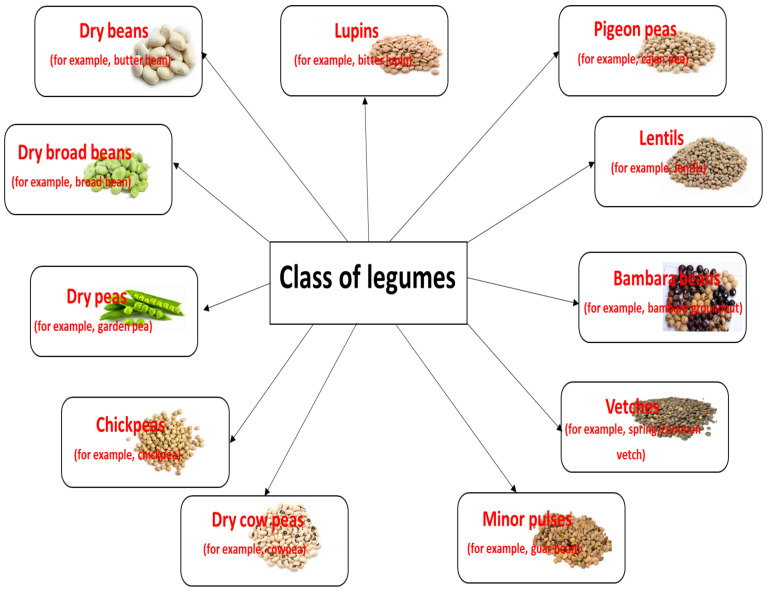
The classification of legumes and their various examples.

**Figure 2 ijms-26-01820-f002:**
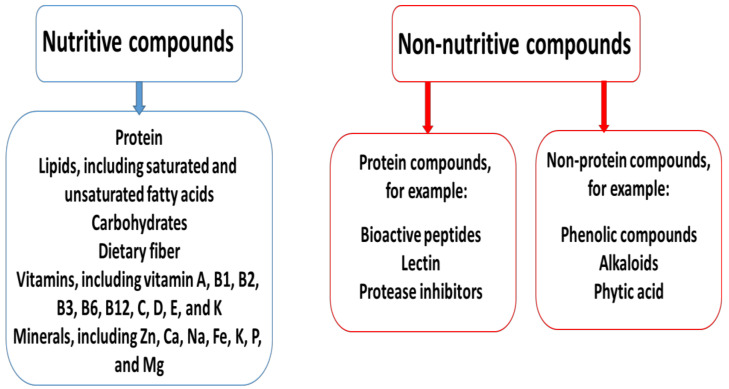
The nutritive and non-nutritive compounds of legumes.

**Figure 3 ijms-26-01820-f003:**
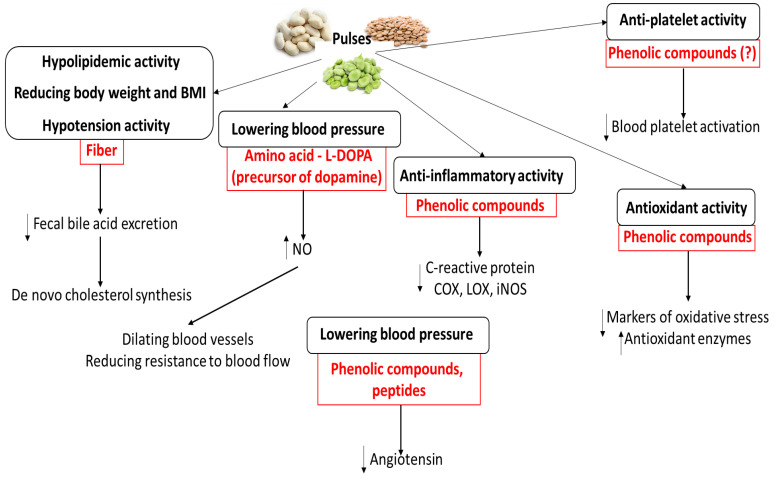
Potential molecular mechanisms of cardioprotection induced by various chemical compounds from pulses.

**Table 1 ijms-26-01820-t001:** Main components of the most commonly known and frequently consumed pulses ([41]; modified).

Pulses	Protein (%)	Lipids (%)	Carbohydrates (%)	Fiber (%)
Chickpea	15.5–28.2	3.1–7.0	44.4	9.0
Broad bean	26.1–38.0	1.1–2.5	37–45.6	7.5–13.1
Kidney bean	20.9–27.8	0.9–2.4	41.5	10
Lentil	23–32	0.8–2.0	46.0	12.0
Pea	18.3–31.0	0.6–5.5	45.0	12.0

(seed weight percentage).

**Table 2 ijms-26-01820-t002:** The effects of pulse consumption on CVD risk factors (clinical trials).

Controlled and Randomized Trials				
**Pulse**	**Dose (g/Day)**	**N**	Effects on CVD Risk Factors	Reference
Beans	50	16 (hypercholesterolemic adults)	Hypolipidemic activity	[33]
Beans	50	23 (hypercholesterolemic adults)	Hypolipidemic activity	[34]
Beans	130	40 (healthy adults)	Hypolipidemic activity	[32]
Beans	130	40 (healthy adults)	Hypolipidemic activity	[32]
Beans	250	28 (healthy subjects)	Hypolipidemic activity	[35]
Beans	140	82 (overweight men and women)	Reduced body weight and BMI	[31]
Beans/lentils	225	123 (obese subjects)	Hypolipidemic activity	[36]
Mixed	160–235	43 (overweight and obese adults)	Hypolipidemic and anti-inflammatory activity	[82]
Mixed	196	50 (subjects with type 2 diabetes)	Hypolipidemic activity	[48]
Mixed	128	40 (overweight and obese adults)	Hypolipidemic activity	[89]
Mixed	442	46 (mature women)	Hypolipidemic activity	[37]
Mixed	250	64 (men with colorectal cancer)	Anti-inflammatory effect	[96]
Pea flour	138	23 (hypercholesterolemic adults and overweight subjects)	Hypolipidemic activity	[85]
Chickpea flour	140	47 (healthy adults)	Hypolipidemic activity	[82]
Chickpea flour	140	27 (healthy adults)	Hypolipidemic activity	[83]
Mixed flour	250	87 (healthy adults)	Hypolipidemic activity	[97]

**Table 3 ijms-26-01820-t003:** Key functional ingredients with cardioprotective activity present in pulses.

Cardioprotective Activity	Responsible Component(s)	References
Lowering blood pressure	Bioactive peptides, phenolic compounds, saponins, fiber	[21,22,50,92,93,94,95]
Hypolipidemic activity	Phytosterols, soluble non-starch polysaccharides, saponins	[10,22,41,45,46]
Anti-inflammatory activity	Phytosterols, phenolic compounds	[112,120,121]
Antioxidant activity	Phenolic compounds, vitamins A and E, bioactive peptides, saponins	[5,9,12,20,22,23,25,26,27,28,29,30,47,48,96,101,102,103,104]
Anti-platelet activity	Phenolic compounds	[114]
Weight reduction and weight control	Bioactive peptides, fiber	[21,115,116,117,118,119]

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
