# Peer review of "The Cardioprotective Properties of Pulses and the Molecular Mechanisms of Their Action"

_ijms, 2025, doi:10.3390/ijms26051820_

Round 1
Reviewer 1 Report
Comments and Suggestions for Authors
This review evaluated the cardioprotective effects of pulses intake on human health with suggestions of some mechanisms of action of selected compounds. The findings are interesting but there are some concerns that should be carefully analyzed by authors:
1) In Title: cardioprotective components change to cardioprotective properties;
2) The author should include a flow diagram illustrating the identification and selection of studies (specific criteria for included and rejected studies and others key information);
3) The effect of pulses consumption on cardiovascular disease is unclear and limited due to low consumption among the studied populations (weak/moderate evidence). In general, high consumption of legumes and pulses has shown beneficial effects on cardiometabolic risk factors but also when associated with a lower intake of ultra processed foods and animal saturated fat. The author should better discuss these questions;
4) The author should remove of mechanisms of action of pulses the word “in vitro” since some items have indication of animal studies;
5) The inclusion of some animal studies about pulse effects on gut microbiota and adiposity induced by obesogenic diets will be interesting for this review;
6) I feel that the review doesn't add new knowledge to existing review literature (https://doi.org/10.1016/j.numecd.2022.10.006; https://doi.org/10.3390/nu16101435; https://doi.org/10.1080/10408398.2020.1716680).
Author Response
This review evaluated the cardioprotective effects of pulses intake on human health with suggestions of some mechanisms of action of selected compounds. The findings are interesting but there are some concerns that should be carefully analyzed by authors:
Response: Thank you for your helpful comments. All of them have been taken into consideration when revising the manuscript. We have added this information in text of manuscript (in red).
1) In Title: cardioprotective components change to cardioprotective properties;
Response: I have changed the title of my manuscript. Now, it is: “The cardioprotective properties of pulses and the molecular mechanisms of their action”.
2) The author should include a flow diagram illustrating the identification and selection of studies (specific criteria for included and rejected studies and others key information);
Response: Because, this manuscript is not systematic review, I have not decided to add a flow diagram illustrating the identification and selection of studies. The chapter – “Research methods” includes this information: “ScienceDirect, PubMed, SCOPUS, Web of Knowledge, Sci Finder, and Web of Science were searched for papers examining functional ingredients with cardioprotective potential present in pulses. The following search terms were used: “legume seeds”; “pulses”, “beans”, “peas”, “broad beans”, “chickpea”, “lentils”, “cardiovascular system”, and “cardiovascular disease” or combinations of the terms. No time criteria were applied to the search, but recent papers were evaluated first. The last search was run on November 15, 2024. The identified articles were first screened by reading the abstract. The molecular mechanisms underpinning the cardioprotective action of pulses was also analyzed and discussed as part of the search.”
3) The effect of pulses consumption on cardiovascular disease is unclear and limited due to low consumption among the studied populations (weak/moderate evidence). In general, high consumption of legumes and pulses has shown beneficial effects on cardiometabolic risk factors but also when associated with a lower intake of ultra processed foods and animal saturated fat. The author should better discuss these questions;
Response: I have added more information about it, especially in the chapter: Meta-analyses and randomized controlled trials. For example: “Additional clinical studies of the effect of pulse consumption on CVD risk factors are summarized in Table 2. Their findings confirm that consumption of pulses and their products has a cardioprotective effect in various models, including inter alia healthy subjects, overweight and obese subjects and hypercholesterolemic adults. In particular, they associate consumption with changes in LDL and HDL cholesterol, and blood pressure. However, a systematic review and dose-response meta-analysis of Mendes et al. (2023) indicate that the intake of legumes is not associated with a reduced risk of stroke. On the other hand, authors suggest that an intake level of 400 g/week seems to provide the optimal cardiovascular benefit. In addition, high consumption of legumes and pulses has shown beneficial effects on cardiometabolic risk factors but also when associated with a lower intake of ultra processed foods and animal saturated fat (Guasch-Ferre et al., 2018; Schwingshackl et al., 2018).”.
4) The author should remove of mechanisms of action of pulses the word “in vitro” since some items have indication of animal studies;
Response: I have not removed “in vitro”, but I have decided to add “animal models” or “human model”: “The control of body weight– in vitro, animal and human models”, and “Anti-inflammatory activity – in vitro and animal models”
5) The inclusion of some animal studies about pulse effects on gut microbiota and adiposity induced by obesogenic diets will be interesting for this review;
Response: I have added more information about it. For example: “Various experiments in animal models also indicate that legume fiber has an anti-obesity effect (117-119). For example, chickpea fiber significantly reduced rat body weight by 14.54% (117), and Wang et al. (118) achieved a similar effect in mice. Wang et al. (118) observed that the consumption of ultrafine grinded pea insoluble dietary fiber increases the abundance of beneficial bacteria (Leptospirosis and Lactobacillus) while reducing the relative abundance of harmful bacteria, including Helicobacter. This successful modulation of microbiota composition increases the production of short-chain fatty acids, resulting in augmented secretion of satiety hormones and presenting the potential for obesity inter-vention. In addition, Lactobacillus, through the production of bile acid hydrolase, has a no-table influence on the metabolism of blood cholesterol by the liver, thereby impacting the dynamics of the bile acid cycle. It has been proposed that this may act by regulating lipid metabolism and influencing gut microbiota structure. In this mice experiment, ultrafine grinded pea insoluble dietary fiber was administered intragastrically one a day at the volume of 10 mL/kg b.w. (0.9 g pea fiber was mixed in 10 mL water), for four weeks.
Recently, the results of Lutsiv et al. (2024) performed on female dietary-induced obesity model mice (C57BL/6J mice) indicate that a mixed pulses (a daily dose of at least 300 g of cooked pulse; for 2 weeks) may to maximize the diversity of the gut microbiome and that the benefits involve the metabolism of vitamins (B1, B6, B9, B12, K2), amino acids (lysine, tryptophan, cysteine), short-chain fatty acids (butyrate, acetate), and other mediators of lipid metabolism. Their menu plan was comprised of a combination of five cooked pulses: dry beans, chickpeas, cowpeas, dry peas, and lentils.
Kadyan et al. (2023) also observed the prebiotic effects of pulses-derive resistant starches on the gut microbiome and intestinal health in aged mice colonized with human microbiota. However, improvements in analytical strategies for the identification of prebiotic saccharides from pulses are needed.
Various systematic reviews (Marinangeli et al., 2020; John et al., 2023) describe that types of pulses and their ingredients can modulate microbial population in the human gut. However, the results of this review show that a limited number of human interventions have studied the effects of a small number of pulses types and pulse ingredients on the microbial population that inhabit the human gut. For example, some studies showed that whole pulses, including chickpeas, cooked navy bean powder did affect the abundance, diversity, and/or richness of gut microbiota in healthy adults (Fernando et al., 2010) and in colorectal cancer survivors (Sheflin et al., 2017). Finley et al. (2007) observed the same effect after the consumption of 130 g canned pinto beans/day (for 12 weeks) in healthy adults and participants with pre-metabolic syndrome. Recently, more details regarding gut microbiota benefits of pulses-based dietary fibers have been described by Biscarrat et al. (2024).”.
6) I feel that the review doesn't add new knowledge to existing review literature (https://doi.org/10.1016/j.numecd.2022.10.006; https://doi.org/10.3390/nu16101435; https://doi.org/10.1080/10408398.2020.1716680).
Response: I have added information from these manuscript in two chapter: “Introduction”, and “Pulses and cardiovascular diseases” – “Meta-analyses and randomized controlled trials”. For example, “Additional clinical studies of the effect of pulse consumption on CVD risk factors are summarized in Table 2. Their findings confirm that consumption of pulses and their products has a cardioprotective effect in various models, including inter alia healthy sub-jects, overweight and obese subjects and hypercholesterolemic adults. In particular, they associate consumption with changes in LDL and HDL cholesterol, and blood pressure. However, a systematic review and dose-response meta-analysis of Mendes et al. (2023) indicate that the intake of legumes is not associated with a reduced risk of stroke. On the other hand, authors suggest that an intake level of 400 g/week seems to provide the optimal cardiovascular benefit.”
“Various studies have also associated regular consumption of pulses with various health benefits (31-37). For example, a systematic review of randomized controlled trials found that a diet rich in pulses has beneficial action on human health, indicating that daily consumption of 54 – 360 g cooked pulses to have cardioprotective benefits (38,39-Ferreira et al., 2021).”
Reviewer 2 Report
Comments and Suggestions for Authors
The manuscript of cardioprotective molecular action mechanisms of pulses components is quite an interesting study. But, the significant gap, as processing is a critical factor in the bioavailability of nutrients and bioactive compounds not well explained. The author described findings but are treated as very incomprehensible to the reader. Like, the generalized findings across different types of pulses (e.g., lentils, chickpeas, beans) without adequately addressing the variability in their bioactive compounds and mechanisms of action. Such manuscripts should be provided with a detailed description of the search sources action mechanisms, which should take into account their advantages, disadvantages, and limitations. In the manuscript, it limits the ability to draw definitive conclusions about the cardioprotective mechanisms of pulses, as the findings are based on secondary data. Some sections, such as the discussion on phenolic compounds and their antioxidant properties and anti-inflammatory activities of pulses, are not detailed. The manuscript briefly mentions that processing techniques (like boiling, canning…., etc.) that can alter the bioactive compounds in pulses, but it does not provide a thorough analysis of how different processing methods impact the cardioprotective properties. In addition, several cited animal and in vitro studies with lack of robust human clinical trials to support the cardioprotective claims. I also suggest the author to include a more detailed discussion of how different processing methods (e.g., boiling, canning, and fermentation) affect the bioactive compounds in pulses and their cardioprotective properties. This could lead to oversimplification of the results. The manuscript briefly mentions the potential role of gut microbiota in the cardioprotective effects of pulses but does not explore this topic in depth. Given the growing interest in the gut-heart axis, this is a missed opportunity to provide a more comprehensive review. The manuscript states that no adverse effects were found in subjects treated with pulses, but it does not discuss potential negative effects, such as gastrointestinal discomfort or antinutritional factors (e.g., phytic acid, lectins). A balanced discussion of both benefits and risks would improve the manuscript. Several typos should be considered like "Desire their variety" to be "Despite their variety."; "Any relevant identified articles were summarized" – This sentence is confused; "an improperly balanced diet" to be "an unbalanced diet."; "It is an important to note" should be "It is important to note; "recruite" should be "recruit.". The manuscript repeats certain points multiple times, such as the cardioprotective effects of phenolic compounds and the role of fiber in reducing cholesterol. This repetition could be reduced to make the manuscript more concise. While the manuscript includes tables and figures, additional visual aids, such as diagrams or flowcharts summarizing the molecular mechanisms of cardioprotection, could enhance understanding.
Comments on the Quality of English LanguagePlease address the grammatical errors and typos identified above to enhance readability.
Author Response
The manuscript of cardioprotective molecular action mechanisms of pulses components is quite an interesting study.
Response: Thank you for your helpful comments. All of them have been taken into consideration when revising the manuscript. We have added this information in text of manuscript (in red).
But, the significant gap, as processing is a critical factor in the bioavailability of nutrients and bioactive compounds not well explained. The author described findings but are treated as very incomprehensible to the reader.
Response: I have added more information about it. For example: “The chemical composition, quality, and biological properties of pulses may be altered by the choice of domestic and industrial food processing technique. Common processing techniques include boiling, soaking, blanching, germination, and micro-wave cooking (54). The method of preparation and thermal processing influences the content of nutritive compounds and non-nutritive compounds. For example, canning and roasting has been found to reduce total phenolic content and antioxidant proper-ties (55), while boiling increases the caroteinoid and tocopherol content in lentils (56) and chickpeas (57). Cooking and fermentation have a beneficial effect by reducing the levels of lectins, which are increased by roasting and baking. On the other hand, phytate content is reduced by cooking, soaking, sprouting, and fermentation (Grdeń and Jakubczyk, 2023). The choice of processing technique can also influence the functional properties of pulse fibers. For example, lentil flours have greater amounts of a-galactoside (a beneficial prebiotic fibre), and less phytic acid after extrusion (58). In another study, cooking led to a significant decrease in resistant starch content in cook beans. Authors suggest that this reduction is attributed to the destruction of amylase inhibitors during the cooking process (Wang et al., 2010). In addition, boiling kidney beans, chickpeas, and white beans increase their total fiber content (Margier et al., 2018). Erba et al. (2019) suggest that germination does not alter the composition of digestive and resistant starch.
Thermal processing can decrease the concentration of certain antinutritional compounds, improving the bioavailability of other food components, and release min-erals from the food matrix (15). Amoah et al. (54) report that the choice of processing technique can improve the bioavailability of nutrients in faba beans which were inhib-ited by anti-nutritional factors. More details about the thermal processing of edible seeds are described in another review paper (15). This review describes that thermal processing of edible seeds, including pulses affects seeds in many ways. Thermal pro-cessing can increase or decrease their biological properties. For example, only one pa-per demonstrates that roasting faba beans (150◦C, 10-120 min) decreases their antiox-idant capacity (Siah, Konczak et al., 2014). Therefore, further experimental testing is needed to identify the precise effects of thermal processing and other preparation methods on the beneficial effects of pulses, including their cardioprotective properties. Such studies should also aim to determine the optimal method, temperature and time of processing.
Like, the generalized findings across different types of pulses (e.g., lentils, chickpeas, beans) without adequately addressing the variability in their bioactive compounds and mechanisms of action. Such manuscripts should be provided with a detailed description of the search sources action mechanisms, which should take into account their advantages, disadvantages, and limitations.
Response: It is difficult to demonstrate cardioprotective properties of different types of pulses. However, it may suppose that they have similar bioactive compounds, which may decide about their cardioprotective potential. In addition, mix of bioactive compounds often decide about biological activity.
I have added more information about in the chapter of Conclusion: “However, the biological mechanisms by which pulse consumption influences the cardiovascular system and CVDs in humans remains unclear and poorly defined. Even so, various chemical compounds (especially, phenolic compounds, fiber, and bioactive peptides) from pulses have been found to show cardioprotective potential; these are summarized in Figure 3, together with their mechanisms. The key functional ingredients of pulses with known cardioprotective activity are given in Table 3. For example, phenolic compounds exert various effects on the body, including anti-inflammatory activity, by regulating signaling pathways such as COX, LOX, and NOS. In addition, the cardioprotective action of phenolic compounds from pulses seems to be associated with their antioxidative activity at the cellular level; they have been found to stimulate antioxidant enzymes. Phenolic compounds may also be responsible for the anti-platelet action of pulses, but their molecular mechanism needs further study.
Recently, Feng et al. (2024) have described that in faba beans L-DOPA (L-3,4-dihydroxy-phenylalanine (amino acid, 1.95 – 2.60 mg/g dry weight) as a precursor to dopamine may regulate blood pressure and help maintain cardiovascular health. L-DOPA may lower blood pressure levels through its ability to stimulate the production of nitric oxide.
Future studies should also investigate the effect of pulse consumption on the human gut microbiota, an important factor in reducing the risk of CVD; while a few papers have found the fiber derived from legumes and their seeds to have the advantageous effects on human health, including dyslipidemia and obesity (41), this area generally remains poorly understood.
It is important to note that no adverse effects were found in any of subjects treated with lentils, pulses or their products. However, more animal studies and clinical trials should be performed to determine their long-term effects and safety in vivo; the latter should aim to recruit both healthy participants, and those with risk factors for CVDs, such as high total and LDL cholesterol and obesity, or who smoke. They should also aim to clarify the effects of processing on the cardioprotective properties of pulses, as this area is currently poorly described.”.
In addition, I have modified Figure 3, which describes potential molecular mechanisms of cardioprotection of pulses.
In the manuscript, it limits the ability to draw definitive conclusions about the cardioprotective mechanisms of pulses, as the findings are based on secondary data. Some sections, such as the discussion on phenolic compounds and their antioxidant properties and anti-inflammatory activities of pulses, are not detailed.
Response: I have added more information about it (the chapter - Meta-analyses and randomized controlled trials). For example: “A key part in the development of CVD is played by inflammation (89). Fortunately, various studies indicate that pulses have anti-inflammatory properties (90,91, North et al., 2009; Haghighatdoost et al., 2017). For example, Hartman et al. (90) report that consumption of navy, pinto, kidney and black beans (about 250 g/day – 1.5 cups) for four weeks reduces the level of inflammation biomarkers, including C-reactive protein, and soluble tumor necrosis factor-a receptors I and II in men with colorectal cancer. A meta-analysis by Salehi-Abargauei et al. (91) also demonstrated a reduction of circulating C-reactive protein (CRP) after consumption of non-soy legumes (N=64).”
Moreover, I have added more information about it in other chapters: “The inhibition of oxidative stress – in vitro”, and “Anti-inflammatory activity – in vitro and animal models”. For example: “Various in vitro studies have found pulses to have anti-inflammatory activity, which is believed to involve various pathways including the inhibition of cyclooxygenase (COX) or lipoxygenase (LOX). For example, Sibul et al. (120) observed that extracts from legumes such as common bean, chickpeas and peas can inhibit COX and LOX, although the precise nature of this activity depends on the type of legumes. Also, Contreras et al. (121) indicate that water-ethanol black bean extract inhibits COX and iNOS activity by 30.6% and 32.3%, respectively. Lee et al. (112) also observed lentils to have anti-inflammatory action (a decrease in proinflammatory cytokines and an increase in anti-inflammatory cytokines (especially IL-10) and renal protective effects for ischemia-reperfusion injury in mice. Recently, Alexander et al. (2014) have described more information about anti-inflammatory effects of lentils.”.
The manuscript briefly mentions that processing techniques (like boiling, canning…., etc.) that can alter the bioactive compounds in pulses, but it does not provide a thorough analysis of how different processing methods impact the cardioprotective properties. In addition, several cited animal and in vitro studies with lack of robust human clinical trials to support the cardioprotective claims. I also suggest the author to include a more detailed discussion of how different processing methods (e.g., boiling, canning, and fermentation) affect the bioactive compounds in pulses and their cardioprotective properties. This could lead to oversimplification of the results.
Response: I have added more information about it. For example: : “The chemical composition, quality, and biological properties of pulses may be altered by the choice of domestic and industrial food processing technique. Common processing techniques include boiling, soaking, blanching, germination, and micro-wave cooking (54). The method of preparation and thermal processing influences the content of nutritive compounds and non-nutritive compounds. For example, canning and roasting has been found to reduce total phenolic content and antioxidant proper-ties (55), while boiling increases the caroteinoid and tocopherol content in lentils (56) and chickpeas (57). Cooking and fermentation have a beneficial effect by reducing the levels of lectins, which are increased by roasting and baking. On the other hand, phytate content is reduced by cooking, soaking, sprouting, and fermentation (Grdeń and Jakubczyk, 2023). The choice of processing technique can also influence the functional properties of pulse fibers. For example, lentil flours have greater amounts of a-galactoside (a beneficial prebiotic fibre), and less phytic acid after extrusion (58). In another study, cooking led to a significant decrease in resistant starch content in cook beans. Authors suggest that this reduction is attributed to the destruction of amylase inhibitors during the cooking process (Wang et al., 2010). In addition, boiling kidney beans, chickpeas, and white beans increase their total fiber content (Margier et al., 2018). Erba et al. (2019) suggest that germination does not alter the composition of digestive and resistant starch.
Thermal processing can decrease the concentration of certain antinutritional compounds, improving the bioavailability of other food components, and release min-erals from the food matrix (15). Amoah et al. (54) report that the choice of processing technique can improve the bioavailability of nutrients in faba beans which were inhib-ited by anti-nutritional factors. More details about the thermal processing of edible seeds are described in another review paper (15). This review describes that thermal processing of edible seeds, including pulses affects seeds in many ways. Thermal processing can increase or decrease their biological properties. For example, only one paper demonstrates that roasting faba beans (150◦C, 10-120 min) decreases their antioxidant capacity (Siah, Konczak et al., 2014). Therefore, further experimental testing is needed to identify the precise effects of thermal processing and other preparation methods on the beneficial effects of pulses, including their cardioprotective properties. Such studies should also aim to determine the optimal method, temperature and time of processing.
The manuscript briefly mentions the potential role of gut microbiota in the cardioprotective effects of pulses but does not explore this topic in depth. Given the growing interest in the gut-heart axis, this is a missed opportunity to provide a more comprehensive review.
Response: I have added more information about it. For example: “Various experiments in animal models also indicate that legume fiber has an anti-obesity effect (117-119). For example, chickpea fiber significantly reduced rat body weight by 14.54% (117), and Wang et al. (118) achieved a similar effect in mice. Wang et al. (118) observed that the consumption of ultrafine grinded pea insoluble dietary fiber increases the abundance of beneficial bacteria (Leptospirosis and Lactobacillus) while reducing the relative abundance of harmful bacteria, including Helicobacter. This successful modulation of microbiota composition increases the production of short-chain fatty acids, resulting in augmented secretion of satiety hormones and presenting the potential for obesity intervention. In addition, Lactobacillus, through the production of bile acid hydrolase, has a no-table influence on the metabolism of blood cholesterol by the liver, thereby impacting the dynamics of the bile acid cycle. It has been proposed that this may act by regulating lipid metabolism and influencing gut microbiota structure. In this mice experiment, ultrafine grinded pea insoluble dietary fiber was administered intragastrically one a day at the volume of 10 mL/kg b.w. (0.9 g pea fiber was mixed in 10 mL water), for four weeks.
Recently, the results of Lutsiv et al. (2024) performed on female dietary-induced obesity model mice (C57BL/6J mice) indicate that a mixed pulses (a daily dose of at least 300 g of cooked pulse; for 2 weeks) may to maximize the diversity of the gut microbiome and that the benefits involve the metabolism of vitamins (B1, B6, B9, B12, K2), amino acids (lysine, tryptophan, cysteine), short-chain fatty acids (butyrate, acetate), and other mediators of lipid metabolism. Their menu plan was comprised of a combination of five cooked pulses: dry beans, chickpeas, cowpeas, dry peas, and lentils.
Kadyan et al. (2023) also observed the prebiotic effects of pulses-derive resistant starches on the gut microbiome and intestinal health in aged mice colonized with human microbiota. However, improvements in analytical strategies for the identification of prebiotic saccharides from pulses are needed.
Various systematic reviews (Marinangeli et al., 2020; John et al., 2023) describe that types of pulses and their ingredients can modulate microbial population in the human gut. However, the results of this review show that a limited number of human interventions have studied the effects of a small number of pulses types and pulse ingredients on the microbial population that inhabit the human gut. For example, some studies showed that whole pulses, including chickpeas, cooked navy bean powder did affect the abundance, diversity, and/or richness of gut microbiota in healthy adults (Fernando et al., 2010) and in colorectal cancer survivors (Sheflin et al., 2017). Finley et al. (2007) observed the same effect after the consumption of 130 g canned pinto beans/day (for 12 weeks) in healthy adults and participants with pre-metabolic syndrome. Recently, more details regarding gut microbiota benefits of pulses-based dietary fibers have been described by Bis-carrat et al. (2024).”.
The manuscript states that no adverse effects were found in subjects treated with pulses, but it does not discuss potential negative effects, such as gastrointestinal discomfort or antinutritional factors (e.g., phytic acid, lectins). A balanced discussion of both benefits and risks would improve the manuscript.
Response: I have added more information about it. For example: “The most commonly-known and frequently-consumed pulses worldwide are lentils (Lens culinaris L.), peas (Pisum sativum L.), chickpeas (Cicer arietinum L.) and beans (Phaseolus vulgaris); together, their production constitutes approximately 70% of the estimated total global output of pulses: lentils (7%), beans (30%), peas (15%), and chickpeas (17%) (Padhi and Dan Ramdath, 2017). However, their consumption is rather low, i.e. below 3.5 kg per capita per year (2). This low consumption has been attributed to gastrointestinal problems and the long cooking time, which can be over 70 minutes (7,11,12), although lentils only require about 25 minutes to cook, making them convenient for human consumption (13). Despite this, in developed countries pulses are utilized more for animal feed (14).”
“The method of preparation and thermal processing also influences the content of nutritive compounds and non-nutritive compounds. For example, cooking and fer-mentation have a beneficial effect by reducing the levels of lectins, which are increased by roasting and baking. On the other hand, phytate content is reduced by cooking, soaking, sprouting, and fermentation (Grdeń and Jakubczyk, 2023). In another study, cooking led to a significant decrease in resistant starch content in cook beans. Authors suggest that this reduction is attributed to the destruction of amylase inhibitors during the cooking process (Wang et al., 2010). In addition, boiling kidney beans, chickpeas, and white beans increase their total fiber content (Margier et al., 2018). Erba et al. (2019) suggest that germination does not alter the composition of digestive and resistant starch.”
Several typos should be considered like "Desire their variety" to be "Despite their variety."; "Any relevant identified articles were summarized" – This sentence is confused; "an improperly balanced diet" to be "an unbalanced diet."; "It is an important to note" should be "It is important to note; "recruite" should be "recruit.".
Response: I have corrected.
The manuscript repeats certain points multiple times, such as the cardioprotective effects of phenolic compounds and the role of fiber in reducing cholesterol. This repetition could be reduced to make the manuscript more concise.
Response: I have corrected.
While the manuscript includes tables and figures, additional visual aids, such as diagrams or flowcharts summarizing the molecular mechanisms of cardioprotection, could enhance understanding.
Response: I have modified Figure 3, which describes potential molecular mechanisms of cardioprotection of pulses.
Round 2
Reviewer 1 Report
Comments and Suggestions for Authors
The author has adequately addressed all issues raised by this reviewer and the revised manuscript is now suitable for publication in IJMS.
Reviewer 2 Report
Comments and Suggestions for Authors